# Chronic Stress That Changed Intestinal Permeability and Induced Inflammation Was Restored by Estrogen

**DOI:** 10.3390/ijms241612822

**Published:** 2023-08-15

**Authors:** Yuanyuan Li, Huayun Wan, Ruiqin Ma, Tianya Liu, Yaoxing Chen, Yulan Dong

**Affiliations:** 1Department of Basic Veterinary Medicine, College of Veterinary Medicine, China Agricultural University, Beijing 100193, China; lyydyxzz@163.com (Y.L.); wanhuayun5012@163.com (H.W.); maruiqin@cau.edu.cn (R.M.); liutianya@cau.edu.cn (T.L.); yxchen@cau.edu.cn (Y.C.); 2Key Laboratory of Precision Nutrition and Food Quality, Ministry of Education, Department of Nutrition and Health, China Agricultural University, Beijing 100193, China

**Keywords:** chronic stress, pregnant mice, intestinal mucosal barrier, microbiota composition, E_2_

## Abstract

Chronic psychological stress affects the health of humans and animals (especially females or pregnant bodies). In this study, a stress-induced model was established by placing eight-week-old female and pregnant mice in centrifuge tubes for 4 h to determine whether chronic stress affects the intestinal mucosal barrier and microbiota composition of pregnant mice. Compared with the control group, we found that norepinephrine (NE), corticosterone (CORT), and estradiol (E_2_) in plasma increased significantly in the stress group. We then observed a decreased down-regulation of anti-inflammatory cytokines and up-regulation of pro-inflammatory cytokines, which resulted in colonic mucosal injury, including a reduced number of goblet cells, proliferating cell nuclear antigen-positive cells, caspase-3, and expression of tight junction mRNA and protein. Moreover, the diversity and richness of the colonic microbiota decreased in pregnant mice. Bacteroidetes decreased, and pernicious bacteria were markedly increased. At last, we found E_2_ protects the intestinal epithelial cells after H_2_O_2_ treatment. Results suggested that 25 pg/mL E_2_ provides better protection for intestinal barrier after chronic stress, which greatly affected the intestinal mucosal barrier and altered the colonic microbiota composition.

## 1. Introduction

Pregnancy is a unique physiological process in the female body, and the stress response during pregnancy has a significant impact on maternal health, delivery outcome, and fetal development. During pregnancy, pregnancy-specific anxiety and depression, triggered by excessive physiological reactions and perceived stress, can further lead to adverse pregnancy outcomes such as premature birth [1,2]. Studies indicate that among females from different towns, 78% experience mild to moderate prenatal psychological stress, and 6% suffer high levels of stress [3]. In response to these stressors, the body’s systems, including the gut, undergo biological changes to adapt and restore balance. Even though stress is important for people’s rapid reaction to threats, chronic stress is associated with detrimental effects on physical health and adversely affects the immune, neuroendocrine, and central nervous systems, promoting the progression of diseases such as fat [4], cardiovascular disease [5], depression [6], and cancer [7].

As the most important immune organ, the intestine plays a critical role in overall health. A growing body of research has shown that chronic psychological stress impairs immune response, reduces the body’s immunity, increases intestinal permeability, and promotes diseases such as inflammation bowel disease (IBD) and cancer [8,9,10]. Psychological stress has also been identified as an important factor in the development of irritable bowel syndrome, significantly impacting intestinal sensitivity, motility, secretion, and permeability [11].

In addition, more studies have demonstrated that the stress response during pregnancy can impact maternal health and childbirth outcomes during pregnancy [12] and even influence the neural development [13], cognitive development [14], bad temperament, and mental disorders of offspring [15]. Therefore, normal regulation of the intestinal tract during pregnancy plays an important role in maternal health.

One characteristic feature of pregnancy is the high levels of estrogens, particularly estradiol (E_2_) [16]. E_2_ is synthesized abundantly by the placenta during pregnancy, promoting angiogenesis [17], vasodilation [18], and the production of angiogenic and stress factors [19]. Additionally, estrogens are known to alleviate oxidative stress levels and exert antioxidant effects in the body [20,21], influencing the physiology, lipid metabolism, protein synthesis, behavior, and diseases of organs and tissues [22,23]. Nevertheless, the specific effects of E_2_ on the intestinal tract of pregnant mice remain unclear. Therefore, a stress-induced pregnant mice model was established in this paper to explore the effects of chronic stress on the endocrine hormones and the intestines of pregnant mice. Additionally, an in vitro oxidative stress epithelial cell model was cultured to explore the role of estrogen in intestine epithelial survival.

## 2. Results

### 2.1. Effect of Chronic Stress on Pregnant Mice

Stress hormones norepinephrine (NE) and corticosterone (CORT) are the most important indicators of animal stress response. We measured the weight and CORT and NE concentrations of pregnant and normal mice of different ages to verify the establishment of the chronic stress model. The weight loss of NP, P5, P10, P15, and P20 after chronic stress increased by 21.32%~66.27% (*p* = 0~0.038) (Figure 1a). CORT concentration increased remarkably by 12.67%~40.82% (*p* = 0.001~0.044), and NE concentration also increased by 13.61%~125.01% in NP, P5, P10, P15, and P20 groups after chronic stress (*p* = 0.000) (Figure 1b,c). Then, we tested the E_2_ concentration in the plasma and found that its concentration decreased by 11.91%~41.45% (*p* = 0.002~0.032) in all groups after chronic stress (Figure 1d). These data indicated that the pregnant mice developed a stress response, and E_2_ concentration decreased after stress. However, no clear trend was observed across different pregnant stages.

### 2.2. Chronic Stress Broke the Intestine Mucosal Mechanical Barrier

To investigate the effects of chronic stress on the barrier and absorption function of intestinal epithelial cells, the changes in the small intestinal villus height (VH), crypt depth (CD), and the ratio (V/C) were detected. H.E. staining results revealed that the VH of different intestinal segments decreased from duodenum to ileum in the same age group (Figure 1e). After stress treatment, VH in ileum, jejunum, and duodenum reduced compared to the Con group (*p* = 0~0.004) (Figure 1f). The CD has the opposite trend, and increased compared to the Con group in ileum, jejunum, and duodenum (*p* = 0~0.006) (Figure 1g). So, the V/C also decreased in all groups after stress treatment compared with the Con group (*p* = 0~0.012) (Figure 1h). Then, we detected Claudin, Occludin, and ZO-1 expression in different segments of the intestine of different pregnant ages mice. After chronic stress, the mRNA expression of Claudin-1 (*p* = 0–0.022), Occludin (*p* = 0–0.004), and ZO-1 (*p* = 0–0.049) in NP, P5, P10, P15, and P20 groups decreased significantly by 1.15%~93.22% in ileum, jejunum, and duodenum (Figure 2a–d). Similarly, the protein expression of Claudin-1 (*p* = 0–0.039), Occludin (*p* = 0.001–0.038), and ZO-1 (*p* = 0–0.033) has the same trend as mRNA levels after chronic stress (Figure 2e–h).

Additionally, we also investigated the effects of chronic stress on the intestinal epithelial cells by immunohistochemistry in the small intestine. The cleaved caspase-3 was mainly present as brown granules in the cytoplasm of the apical intestinal villus (Figure 3a). The PCNA-positive cells showing brown or brownish yellow granules mostly appeared along the crypt bottom (Figure 3b). Statistically, compared with the same intestinal segment of the normal control group at the same age, the cleaved caspase-3 positive expression in duodenum, jejunum, and ileum increased significantly by 40.94%~132.05% after chronic stress in NP, P5, P10, P15, and P20 (*p* = 0.000) (Figure 3c), while PCNA-positive expression of duodenum, jejunum, and ileum was significantly decreased by 15.99%~52.41% in NP, P5, P10, P15, and P20 (*p* = 0.000). In conclusion, braking stress can significantly reduce the expression of PCNA in female intestinal epithelial cells, and the proliferation is slow (Figure 3d). Data indicate chronic stress affects the intestinal epithelial barrier structure, inhibits tight junction protein expression, and increases intestinal permeability.

### 2.3. Chronic Stress Damaged Intestinal Mucus Layer of Pregnant Mice

Next, we measured the amount of goblet cells and the expression of MUC2 mRNA in order to determine whether chronic stress affects mucus layer thickness in pregnant mice. PAS staining revealed that the number of goblet cells increased from the duodenum to the ileum (Figure 4a). The results showed that after chronic stress, the number of goblet cells in duodenum, jejunum and ileum epithelial cells in NP, P5, P10, P15, and P20 decreased by 10.06%~36.6% (*p* = 0~0.004) (Figure 4b. Compared with the same intestine of the same day female mice of the control group, the number of goblet cells and the expression of MUC2 mRNA decreased dramatically by 26.81%~82.36% (*p* = 0–0.017) in small intestine and colon of the pregnant mice group (Figure 4b–f). In conclusion, chronic stress adversely affects the number of goblet cells and their function in pregnant mice.

### 2.4. Effect of Chronic Stress on Pregnant Mice Affected Intestinal Mucosal Immune Function

To investigate the effect of chronic stress on intestinal mucosal immune function in pregnant mice, we detected the sIgA concentration in intestinal mucosa and intestinal inflammatory cytokines using ELISA and CBA of different day age mice. sIgA plays an important role in intestinal immunity. It is mainly produced by IgA plasma cells in the lamina propria of intestinal mucosa. When intestinal mucosal epithelium is damaged, sIgA secretion is bound to be affected. We found that the sIgA concentration gradually increased from the duodenum to the ileum of NP, P5 (except ileum), P10, P15, and P20 (except ileum) (Figure 5a). Moreover, sIgA concentrations increased by 4.97%~130.62% (*p* = 0–0.032) after chronic stress in all stages during pregnancy (Figure 5a). Furthermore, the concentrations of the pro-inflammatory cytokines IFN-γ (*p* = 0–0.117), TNF-α (*p* = 0–0.021), IL-2 (*p* = 0–0.104), and IL-6 (*p* = 0–0.004) increased significantly by 10.57%~1136.93% in duodenum, jejunum, and ileum after chronic stress in all stages during pregnancy. The anti-inflammatory cytokines IL-4 (*p* = 0–0.034) and IL-13 (*p* = 0–0.008) had an opposite trend and decreased by 13.90%~51.94% (Figure 5b–e). The overall results showed an increase in pro-inflammatory cytokine concentration and a decrease in anti-inflammatory cytokines in the pregnant mice group. These data indicated that chronic stress could promote abnormal activation of intestinal immunity in mice, leading to intestinal inflammation.

### 2.5. Effect of Chronic Stress on Pregnant Mice Altered Gut Microbiota Composition

Finally, we assessed whether chronic stress could affect the gut microbiota in the P10 group. The α-diversity analysis of shannon curve, rarefaction curve, and rank abundance curve showed that the richness and diversity of the colonic microbiota decreased significantly in the P10 group after chronic stress (Appendix A). ACE index, Simpson index, Shannon index, and Chao1 index also showed that chronic stress decreased alpha diversity (Appendix A). Then, species abundance histogram and PCoA analysis presented a distinct difference between con and stress groups (Figure 6a,b). Through LEfSe analysis, we found that the main dominant flora also varied greatly in con and stress groups (Figure 6c,d). Moreover, the relative abundance of *Bacteroides*, *Prevotellaceae*, *Ruminococcaceae*, and *Alistipes* decreased significantly, and *Camobacteriaccae*, *uncultured_bacterium_g_Staphylococcus* increased significantly in the P10 group after chronic stress. These results showed that chronic stress can significantly affect colonic microbiota composition, decreasing the relative abundance of Bacteroidetes and increasing the relative abundance of pathogenic bacteria.

### 2.6. Low Concentration E_2_ Protects Intestinal Epithelial Cells

To further explore the role of estrogen in the intestinal epithelium, we examined how different estrogen concentrations affected cell proliferation, apoptosis, and expression of tight-linked proteins. First, we used H_2_O_2_ processing cells to build an oxidation emergency model that simulated chronic stress. We treated cells with different concentrations of H_2_O_2_ to detect the proliferation of cells after damage. When H_2_O_2_ was found to be 1000 pg/mL, the decrease in cell proliferation activity was the most obvious, and the oxidative stress cell model was successfully established (*p* = 0.000) (Figure 7a). We treated cells with different concentrations of E_2_ and found that as the concentration increased, cell proliferation capacity gradually increased, but at concentrations greater than 50 pg/mL, cell proliferation began to decline (*p* = 0.000) (Figure 7b). We then used different concentrations of E_2_ to treat the IPEC cells of H_2_O_2_ damage and found that when the concentration of E_2_ was 25 pg/mL, the damage caused by H_2_O_2_ to the cells led to a significant increase in proliferation activity (*p* = 0.015) and a significant decrease in the percentage of apoptosis (*p* = 0.002) (Figure 7c,d). 

Then, we tested the effect of different concentrations of E_2_ on IPEC tight junction protein. In comparison with DMSO, Claudin-1 (*p* = 0.000), Occludin (*p* < 0.001), and ZO-1 (*p* = 0.000) decreased significantly after H_2_O_2_ treatment. Compared with the H_2_O_2_ group, within a certain range (0–50 pg/mL), with the increase of E_2_ concentration, the relative expression of Claudin-1, Occludin and ZO-1 protein increased. When E_2_ concentration reached 25 pg/mL, the relative expression of Claudin-1 (*p* = 0.024), Occludin (*p* = 0.050), and ZO-1 (*p* = 0.007) increased significantly (Figure 7e–h). It appears that E_2_ at the appropriate concentration has a protective effect on intestinal epithelial cells, and the protective effect of E_2_ is greatest at a concentration of 25 pg/mL.

## 3. Discussion

Intestinal health is guaranteed when the intestinal barrier is functionally intact and intestinal flora is in balance [24,25]. Maternal health and development are profoundly influenced by the environment and behavior during pregnancy [2]. Research indicates that women experience depression nearly twice as often as men [26,27,28]. Increased stress levels have been associated with premature births [29], and pregnant individuals are more susceptible to depression due to physiological, psychological, and social factors. In addition, stress during pregnancy raises the risk of diseases in future generations, such as schizophrenia [30] and depression [31]. However, it is not known what effect stress has on the maternal intestine during pregnancy. Our results demonstrated that chronic stress impairs the maternal intestinal barrier, increases intestinal permeability, changes flora composition, and causes intestinal inflammation at different stages of pregnancy. Additionally, E_2_ plays a role in modulating the effects of chronic stress on the gut.

In this study, mice were treated with continuous stress. We observed that NE and CORT concentrations in plasma increased significantly in the pregnant group [32], whereas E_2_ concentrations decreased significantly. The main stress hormones, NE and CORT, play a crucial role in neuroendocrine regulation and reflect the stress response. In early studies, estradiol potentiates basal and stimulated levels of adrenocorticotropic hormone and corticosterone [33,34,35]. Low-dose, short-term estradiol administration inhibits HPA axis activity [36,37,38,39], and at higher doses and over long periods, the responses are enhanced [33,34,35,40]. We observed stable estrogen levels in mice in the NP group and significant increases in NE, CORT, and estrogen in the stress group. In the first trimester, estrogen levels begin to increase. Stress activates the HPA axis, leading to decreased estrogen levels, which, in turn, inhibits the HPA axis. Thus, pregnant mice exhibit heightened sensitivity to stress during pregnancy.

The morphology of mucous membranes, including mucosa thickness, crypt depth, villi architecture, and goblet cell counts, is used to evaluate gut barrier function. Stress-related gastrointestinal symptoms include nausea, vomiting, abdominal pain, and changes in bowel habits [11,36,39]. In our study, we observed a significant decrease in villus height (V) and the V/C ratio, as well as the number of goblet cells and expression of MUC2 mRNA, along with a significant increase in crypt depth. These findings align with previous research indicating that stress can disrupt the integrity of the intestinal barrier [41,42,43]. 

The balance in proliferation and apoptosis of intestinal epithelial cells is necessary for maintaining the normal physiological function of intestinal mucosal barrier [44]. In this study, we further analyzed the effects of chronic stress on intestinal epithelial cells in pregnant mice. We found that chronic stress significantly up-regulated the expression of caspase-3, suggesting increased apoptosis. Increased apoptosis of intestinal epithelial cells can lead to damage of intestinal mucosal barrier, leading to serious intestinal injury [45,46]. Conversely, PCNA is an antigen expressed in proliferating nuclei, which is crucial for nuclear DNA synthesis and is closely associated with cell proliferation [47,48]. The down-regulation of PCNA indicated a decrease of cell proliferation. Increased apoptosis and decreased cell proliferation lead to decreased intestinal epithelial cell regeneration. Based on the previous results, stress caused an increase in NE and CORT levels, and excess glucocorticoids have been reported to be able to cause cell damage, causing increased apoptosis, which may be another factor for chronic stress-induced damage to the intestinal mucosal [49,50,51].

Mucosal immunity barrier, which is the body’s first defense against pathogenic invasion and infection, relies on closely arranged epithelial cells and various immune cells, such as intraepithelial lymphocytes, M cells, and other immunes, to maintain the body’s health. In addition, intestinal mucosal epithelial cells can secrete large amounts of mucus, forming a layer of mucus on the surface of these epithelial cells that prevents microorganisms from attaching to epithelial cells [52]. In the study, we observed that chronic stress can significantly affect the distribution of intestinal mucosa sIgA in pregnant mice. Compared to the control group, the sIgA content gradually increased from small intestine to large intestine because the duodenum was rich in food antigens, while the ileum and colon were rich in microorganisms [50]. In addition, we found that the sIgA content of ileum is higher than that in the colon, attributed to the concentrated distribution of lymphoid nodules known as Peyer’s patches (PP) in the ileum [51,52]. B lymphocytes in PP differentiate into plasma cells, the primary source of sIgA. The increased load of intestinal antigen received by PP can stimulate immature B lymphocytes to differentiate into plasma cells and secrete more sIgA. Consequently, compared to the control group, the pregnant group exhibited increased sIgA content in each intestinal segment. By combining these findings with previous experimental results, we speculated that chronic stress may damage the mucosal barrier in each intestinal segment, resulting in in situ infection or translocation of some colonizing bacteria in the intestine, thereby increasing the presence of pathogenic bacteria in the intestine. In response, the intestinal epithelium’s first immune barrier, sIgA secretion, undergoes compensatory enhancement. Studies have found that stress often leads to severe intestinal inflammation by promoting the release of pro-inflammatory cytokines, such as TNF-α, IL-1, IL-8, and IL-6 [53]. Furthermore, Lina Wei detected whether pro-inflammatory cytokines induced intestinal inflammation of CUMS (chronic unpredicted mild stress). The results showed that CUMS treatment significantly increased IL-6 and INF-γ expression in the colon [54]. In our study, chronic stress activates the HPA axis, with increased NE and CORT secretion to enhance immunity. High levels of NE and CORT and further immune cell activity [12] lead to increased secretion of pro-inflammatory factors and subsequent intestinal inflammation. These findings are consistent with our previous results [55,56], supporting the notion that chronic stress plays a significant role in intestinal inflammation.

Moreover, we conducted an analysis of the colonic microbiota composition. The results revealed a decrease in the diversity and richness of intestinal microflora, along with a reduction in the relative abundance of Bacteroides in colonic contents and an increase in the relative abundance of proteus. These changes may be attributed to the decreased expression of MUC2, resulting in increased permeability of the intestinal mucosal barrier, which facilitated pathogen translocation, inhibited Bacteroides proliferation, and promoted Proteus proliferation [57]. Studies have identified that the frequency of Firmicutes and Bacteroidetes is an important indicator of structural modifications of the gut microbiota, and the balance among gut commensal bacteria plays a crucial role in regulating the gut barrier and metabolic functions [24,58]. In addition, Bacteroidetes assist in decomposing polysaccharides, promoting the host immune system development, and maintaining the intestinal microecological balance [59,60]. The shift on these microbial communities may contribute to stress-induced disruption of the intestinal barrier.

Polysaccharides are considered activators of B cells [60], and stress can increase pro-inflammatory factors by influencing B cells. As a result, to suppress overactivated B cells, the proportion of Bacteroides is reduced, and the proportion of microflora is imbalanced. In newborn mice compared with adult mice, the proportion of Proteobacteria decreases significantly with age, while Bacteroides gradually expands [61]. Clinical studies reveal that the developing intestine is more prone to inflammation [62], with early aberrant bacterial colonization believed to play a role in this process. A decrease in the proportion of Bacteroides causes amplification of the proportion of Proteobacteria, which can lead to the destruction of the intestinal barrier and the appearance of inflammation [63]. We also found that the P20 results have no statistical significance, which may be due to the fact that the pregnant mice were about to give birth after pregnant 20d. Severe labor pain can increase sympathetic excitement, stress response, and sensitivity to pain in P20 pregnant mice, resulting in systemic stress, causing a series of neural and endocrine responses and changing various functions and metabolism, so the data of P20 pregnant mice are not statistically significant [64].

Women need adequate nutrition during pregnancy, and the intestinal barrier affects the absorption of intestinal nutrients. Our results suggest that chronic stress leads to inflammation in pregnant mice. During pregnancy, female estrogen levels increase significantly. Immunological, epidemiological, and clinical evidence suggest that female sex hormones play an important role in the etiology and pathophysiology of chronic immune inflammatory diseases [65,66], and estrogen inhibits the apoptosis of immune cells [67,68]. In our study, we treated cells with H_2_O_2_ and E_2_ to explore the effects of estrogen on intestinal epithelial cells. The result was similar with other’s study, showing appropriate concentrations of estrogen promote the proliferation of intestinal epithelial cells, inhibit their apoptosis, and increase the expression of tightly connected proteins. Moreover, women produce higher Th2 reactions and antibodies, providing better protection against infection, but high immune responses can also make them susceptible to autoimmune diseases [66]. Therefore, after chronic stress in pregnant women, high levels of estrogen can further stimulate immune cell activation, causing inflammation [69]. It is similar with our result that some inflammatory factor levels increase significantly in women after pregnancy and further increase after stress. Long-term chronic inflammation causing confusion of the flora, intestinal stability imbalance, intestinal barrier damage, threatening women’s health [29]. linked protein levels of IPEC cells and found that high levels of estrogen even further damaging to the intestinal barrier. Our findings also indicate that high levels of estrogen can further damage the intestinal barrier, as observed in the effects on tightly linked protein levels of IPEC cells.

In summary, chronic stress significantly induces stress responses in maternal mice, leading to reduced estrogen secretion, disruption of the intestinal barrier, increased secretion of pro-inflammatory cytokines, decreased secretion of anti-inflammatory cytokines, and dysbiosis of the gut microbiota. Moreover, 25 pg/mL E_2_ protected the intestine. Our research demonstrates that maintaining stable maternal estrogen levels during pregnancy can safeguard intestinal health. However, we have yet to delve into the specific mechanisms underlying the actions of estrogen and have not fully considered the impacts of changes in other hormone levels. Therefore, further research is promising to delve deeper into the specific mechanisms of estrogen action and comprehensively explore the effects of other hormonal fluctuations in the future.

## 4. Materials and Methods

### 4.1. Animals and Treatment

There were 150 unbred pure ICR mice (8-week-old), 120 females, and 30 males, purchased from Beijing Vital River Laboratory Animal Technology Co., Ltd. (Beijing, China). Animals were housed under standard conditions (temperature: 21 ± 1 ℃, relative humidity: 50 ± 10%) with a regular 14/10 light/dark cycle (with lights on at 7:00 AM). Animals were provided enough food and water for seven days to adapt to the new environment.

Female mice in estrus without pregnancy were named NP (not pregnancy).

At around 16:00 every day, mice were given vaginal smears, stained with toluidine blue, and observed under a microscope. Animals in estrus were selected, and the female mice were identified as estrus. The estrus and male rats were cage together overnight at around 18:00, and the female rats were examined by vaginal suppository at 7:00–8:00 in the morning of the next day. The pregnancy of the female rats with vaginal suppository was determined to be 1d (P1). During P1, P6, P11, and P16 pregnancies, pregnant mice were put into self-made brake tubes for 4 h braking stress at 8:00 am, respectively. The weight, water intake, and food intake of the mice before and after stress were recorded. The mice were subjected to such continuous stress treatment for 5 days until P5, P10, P15, and P20. The samples of intestine and colon were collected. All animal procedures were approved by the China Agricultural University Institutional Animal Care and Use Committee (AW03602202-2-1).

### 4.2. HE and PAS Staining

Intestinal segments were immediately fixed with 4% paraformaldehyde for 48 h and embedded with paraffin (5 μm, cross-section). Hematoxylin and eosin (H&E) and periodic acid–Schiff (PAS) staining was performed on 5 μm sections. These images were obtained on a vertical DP72 microscope (Olympus, Tokyo, Japan).

### 4.3. Immunohistochemical Staining

For the immunohistochemical studies, the sections were incubated overnight at 4 °C with the monoclonal rabbit anti-mouse primary antibody (PCNA, 1/500; Abcam, Cambridge, CA, USA) and the polyclonal cleaved caspase-3 primary antibody (1/200, Abcam, Cambridge, CA, USA). Then, the sections were rinsed with 0.01 mol/L PBS (pH 7.4) and incubated with biotinylated sheep anti–rabbit IgG (1/300; Sigma, Cambridge, CA, USA) for 2 h at room temperature. After washing, the tissues were incubated with streptavidin–horseradish peroxidase (1/300, Sigma, St. Louis, MO, USA) for 2 h and 30 min at room temperature. Immunoreactivity was visualized by incubating the tissue sections in 0.01 mol/L PBS containing 0.05% 3′,3–diaminobenzidine tetrahydrochloride (DAB; Sigma, St. Louis, MO, USA) and 0.003% hydrogen peroxide for 10 min in the dark. The sections were then stained with hematoxylin and mounted. The images were acquired on an upright DP72 microscope (Olympus, Tokyo, Japan). Image analysis system (Image-pro plus 6.0) was used to count the number of PCNA–positive cells and measure the mean integral optical density (IOD) of cleaved caspase-3 positive cells. The data were counted from 10 random fields from three cross-sections of the three specimens from each group.

### 4.4. Western Blotting

We homogenized the intestine and colon tissues and centrifuged them at 12,000× *g* for 10 min at 4 °C, collecting the supernatants. BCA (bicinchoninic acid) kit (Beyotime, Co., Ltd, Shanghai, China) was used to measure protein concentrations. Proteins were boiled at 99 °C for 10 min, electrophoresed in 10% SDS polyacrylamide gel, and transferred to PVDF membranes (Millipore, Burlington, MA, USA). Blots were blocked with 5% skim milk in Tris-buffered saline solution–Tween 0.1% for 3 h at room temperature and incubated with primary antibodies overnight at 4 °C (mouse anti-Claudin-1, 1/1500, abcam, Cambridge, CA, USA; mouse anti-Occludin-1, 1/1000, CUSABIO, Beijing, China; mouse anti-ZO-1, 1/500, PROTEINTECH, Beijing, China; mouse anti-β-actin, 1/10,000, CWBIO, Beijing, China). After washing and incubating with the secondary antibodies (sheep-anti-rabbit/mouse IgG, 1/4000, Solarbio, Beijing, China) for 2 h at room temperature, the blots were developed with enhanced chemiluminescence (WBKLS0500, Millipore, Billerica, MA, USA). The bands on the blots were scanned and measured using ImageJ (version 4.0.2; Scion Corp.). The results are expressed as IODs for the bands, and three repeats were conducted.

### 4.5. Measurement of Plasma NE, CORT, E_2_, sIgA, and Intestinal Inflammatory Factors Concentration

The protein concentrations of the plasma and intestinal segments were determined using bicinchoninic acid (BCA) (mg/mL) prior to testing CORT (LBTR-EL-1490, Beijing Lebo Terri Technology Development Co. Ltd., Beijing, China), NE (LBTR-EL-1487, Beijing Lebo Terri Technology Development Co. Ltd., Beijing, China), and sIgA (LBTR-EL-1537, Beijing Lebo Terri Technology Development Co. Ltd., Beijing, China) concentrations using a competitive ELISA (CEA540Ge, Uscn Life Science Inc., Wuhan, China), and the inflammatory factor concentration using LEGEND 8-plex Mouse TH1/TH2 Panel (740029, Biolegend, San Diego, CA, USA). All tests were performed according to the manufacturer’s instructions. The experiments were repeated two times, and the final concentration was presented in pg/mg protein. E_2_ was detected via radioimmunoassay (RIA). Each sample was tested in triplicate.

### 4.6. RNA Extraction and Quantitative Real-Time Polymerase Chain Reaction

Total RNA was extracted using Trizol reagent (CW0580A, CWBIO, Beijing, China). The concentration and purity of RNA were measured by NanoPhotometer (P330, Implen, Munich, Germany). Afterwards, 2 μg of total RNA was mixed with reverse transcriptase, and other reagents from the cDNA were synthesized using the GoScriptTM Reverse Transcription System (A5001, Promega, Madison, WI, USA). The cDNA was diluted 10 times to conduct polymerase chain reactions. Primers were synthesized by Invitrogen Trading of Shanghai. qPCR was performed using qPCR SYBR Green Master Mix (Vazyme, Nanjing, China). The primer sequences used for these studies are shown in Table 1.

### 4.7. Cell Culture and Treatment

IPEC-J2 cells are intestinal porcine enterocytes isolated from the jejunum of a neonatal unsuckled piglet. They were cultured in DMEM with 100 units/mL penicillin, 100 mg/mL streptomycin, and 10% fetal bovine serum (FBS) (GIBCO, NewYork, NY, USA) in a CO_2_ incubator (5% CO_2_, 37 °C). Before treatment, cells were plated at 10^6^ or 10^4^ cells/well in DMEM with 10% FBS (complete medium) for 6 h. For drug treatment, the cells were cultured at a density of 2 × 10^6^ cells/mL in 6-well plates. The cells were pretreated for 30 min using E_2_ (E2758, Sigma, St. Louis, MO, USA) and H_2_O_2_, followed by H_2_O_2_. This drug treatment was allowed to continue for 24 h. The cells were then frozen and used to detect proliferation by CCK-8.

### 4.8. Cell Counting Kit-8 (CCK-8)

The IPEC-J2 cells were seeded in 96-well plates; approximately 10^4^ cells were seeded per well. After culturing for 24 h at 37 °C in 5% CO_2_, the cells were divided into several groups with different treatments; each group had at least 3 repetitions. Ten microliters of CCK-8 reagent (CA1210, Solarbio, Beijing, China) was added to each well and incubated for 2 h under the above conditions. The absorbance at 450 nm was measured by a microplate reader.

### 4.9. Microbial Sequencing

The microflora detection was completed by Baimaike Company (Beijing, China). Fresh fecal pellets were collected and store in −80 ℃. The first step was to obtain bacterial genomic DNA from frozen colon contents. The amplicon library preparation was performed by polymerase chain reaction (PCR) amplification of the V3–V4 region of the 16S rRNA gene and then performed on Illumina MiSeq 2500. Sequences with ≥97% similarity were assigned to the same operational taxonomy units (OTUs) by QIIME (version 1.8.0) [15]. The community diversity (Simpson) was analyzed using Mothur (version v. 1.30). β-Diversity was calculated based on the principal coordinate analysis (PCoA) to identify microbial differences among samples. A Venn diagram was implemented to show unique and shared OTUs. All 16S rRNA gene sequencing read data have been deposited in the NCBI Sequence Read Archive (SRA) repository under accession number PRJNA991246.

### 4.10. Statistical Analysis

All the data were expressed as the means ± standard deviations (SD) and were analyzed using IBM SPSS Statistic 20 Software (SPSS Sciences, Chicago, IL, USA). Unpaired *t*-tests, ANOVA, and a post hoc least significant difference (LSD) test were performed. Differences with *p* < 0.01 were considered very statistically significant, and *p* < 0.05 was considered significant.

## 5. Conclusions

In summary, chronic psychological stress significantly disturbed the pregnant maternal endocrine system, such as increased NE, CORT, and E_2_ in plasma. Meanwhile, chronic psychological stress resulted in colonic mucosal injury, pro-inflammatory reaction, and decreased the diversity and richness of the colonic microbiota in pregnant mice. It was interesting that 25 pg/mL E_2_ provides better protective effect on intestinal epithelial cells (Figure 8). These results can help us find that women’s intestinal health during pregnancy needs attention and the role of E_2_ in protecting intestinal health, but the specific ways in which E_2_ works require further study.

## Figures and Tables

**Figure 1 ijms-24-12822-f001:**
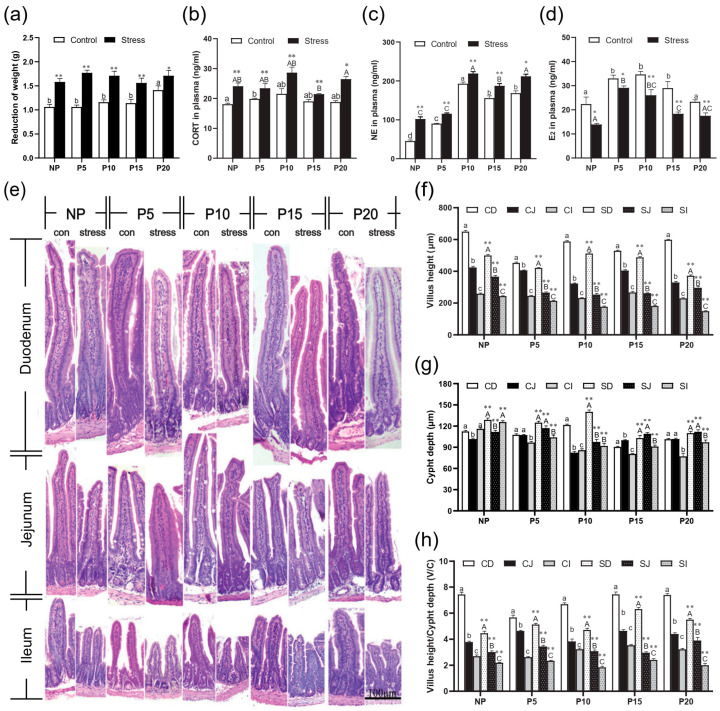
Chronic stress altered weight, CORT and NE in the plasma and histological structure of pregnant mice. (**a**)Effects of chronic stress on the weight of pregnant mice. (**b**–**d**) Effects of chronic stress on the concentration of CORT (**b**), NE (**c**), and E_2_ (**d**) in the plasma of pregnant mice (*n* = 6–8). (**e**) The changes of small intestinal histological structure of pregnant mice (*n* = 4) (H.E. staining, scale bars = 100 μm). (**f**–**h**) Effect of chronic stress on the small intestinal villus height (**f**), crypt depth (**g**), and V/C value (**h**) of pregnant mice (*n* = 4). * *p* < 0.05, ** *p* < 0.01 by Student’s t-test compared with the control group. Letter casing represents the comparison of different segments at the same age by one-way ANOVA. The lowercase letters represent the comparison of different segments of the control groups at the same age, and the uppercase letters represent the comparison of different segments of the stress groups at the same age. The same letter indicates that the difference is not significant (*p* > 0.05), while different letters indicate that the difference is significant (*p* ≤ 0.05). CD: Duodenum of control group; CJ: Jejunum of control group; CI: Ileum of control group; SD: Duodenum of chronic stress group; SJ: Jejunum of chronic stress group; SI: Ileum of chronic stress.

**Figure 2 ijms-24-12822-f002:**
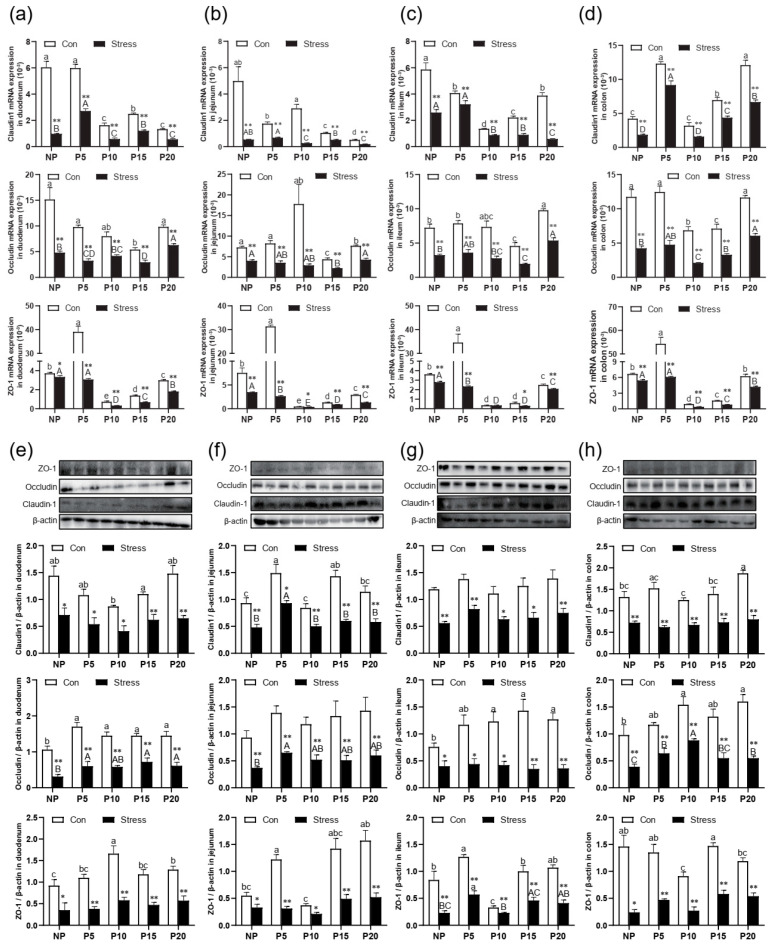
Effect of chronic stress on the relative tight junction mRNA and protein expression level of intestine in pregnant mice. (**a**–**d**) Effect of chronic stress on the relative ZO-1, Occludin, and Claudin-1 mRNA expression level of duodenum (**a**), jejunum (**b**), ileum (**c**), and colon (**d**) in pregnant mice (*n* = 6). (**e**–**h**) The expression of tight junction protein ZO-1, Occludin, Claudin-1, and β-actin protein were examined by western blotting in duodenum (**e**), jejunum (**f**), ileum (**g**), and colon (**h**) after stress in pregnant mice, and relative protein levels were normalized to β-actin (*n* = 3). * *p* < 0.05, ** *p* < 0.01. Letter casing represents the comparison of different segments at the same age by one-way ANOVA. The lowercase letters represent the comparison of different segments of the control groups at the same age, and the uppercase letters represent the comparison of different segments of the stress groups at the same age. The same letter indicates that the difference is not significant (*p* > 0.05), while different letters indicate that the difference is significant (*p* ≤ 0.05).

**Figure 3 ijms-24-12822-f003:**
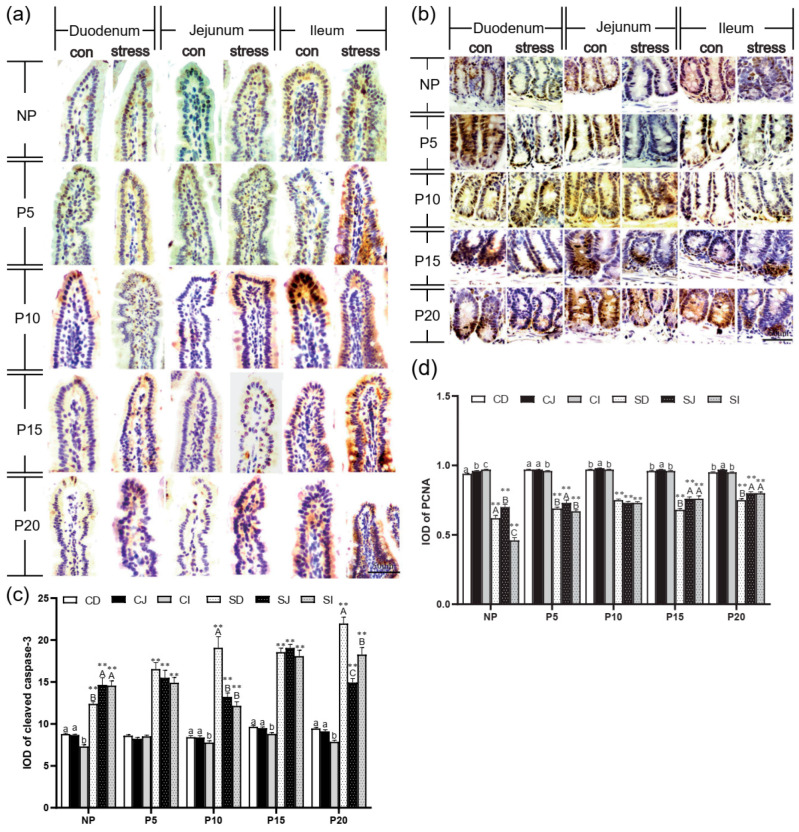
Chronic stress damaged intestinal mucus layer of pregnant mice. (**a**,**b**) Expression and distribution of cleaved caspase-3 (**a**) and PCNA-(**b**) positive cells in the small intestine of pregnant mice (IHC, scale bars = 50 μm). (**c**,**d**) Effect of chronic stress on the hangs of cleaved caspase-3 (**c**) and PCNA (**d**) expression of small intestine in pregnant mice (*n* = 3). ** *p* < 0.01. Letter casing represents the comparison of different segments at the same age by one-way ANOVA. The lowercase letters represent the comparison of different segments of the control groups at the same age, and the uppercase letters represent the comparison of different segments of the stress groups at the same age. The same letter indicates that the difference is not significant (*p* > 0.05), while different letters indicate that the difference is significant (*p* ≤ 0.05). CD: Duodenum of control group; CJ: Jejunum of control group; CI: Ileum of control group; SD: Duodenum of chronic stress group; SJ: Jejunum of chronic stress group; SI: Ileum of chronic stress.

**Figure 4 ijms-24-12822-f004:**
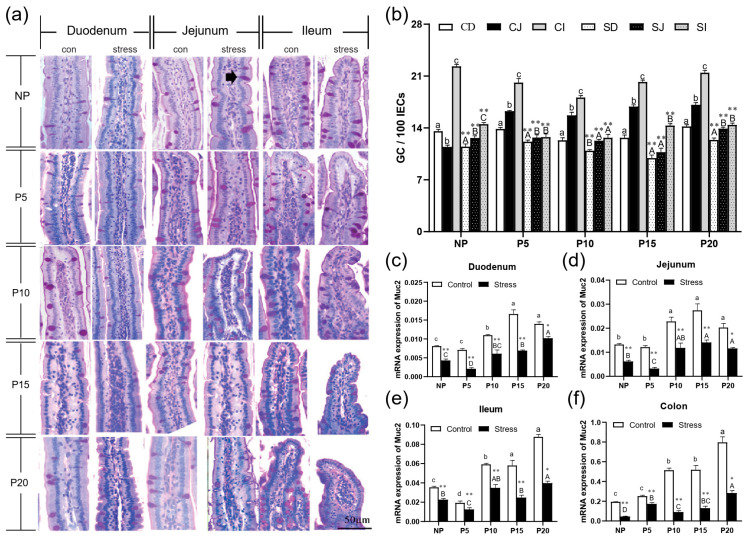
Chronic stress damaged intestinal mucus layer of pregnant mice. (**a**) The distribution of goblet cells in the small intestine of pregnant mice (PAS staining, black arrow points to the goblet cells, scale bars = 50 μm). (**b**)Effect of chronic stress on the number of goblet cells (GC) of small intestine in pregnant mice (*n* = 3–4). (**c**–**f**) Effect of chronic stress on the relative Muc2 mRNA expression level of duodenum (**c**), ileum (**d**), jejunum (**e**), and colon (**f**) of pregnant mice (*n* = 6). * *p* < 0.05, ** *p* < 0.01. Letter casing represents the comparison of different segments at the same age by one-way ANOVA. The lowercase letters represent the comparison of different segments of the control groups at the same age, and the uppercase letters represent the comparison of different segments of the stress groups at the same age. The same letter indicates that the difference is not significant (*p* > 0.05), while different letters indicate that the difference is significant (*p* ≤ 0.05). CD: Duodenum of control group; CJ: Jejunum of control group; CI: Ileum of control group; SD: Duodenum of restraint stress group; SJ: Jejunum of restraint stress group; SI: Ileum of restraint stress group.

**Figure 5 ijms-24-12822-f005:**
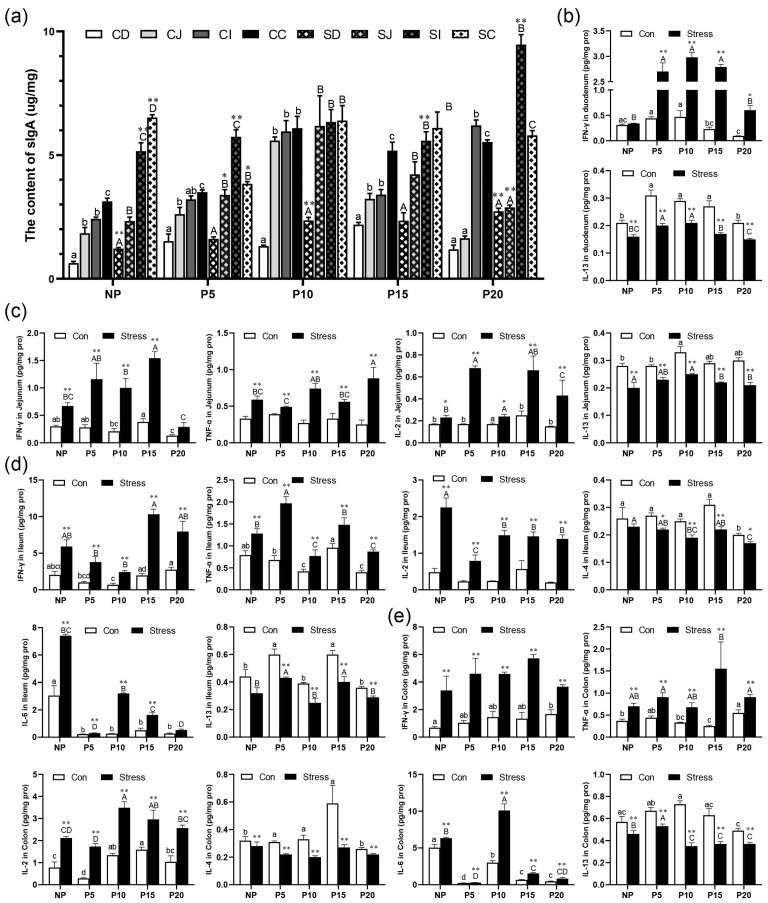
Chronic stress affected intestinal mucosal immune function. (**a**) Effect of chronic stress on the sIgA content of intestine in pregnant mice (*n* = 4–6). (**b**) Effect of chronic stress on the expression level of cytokines (IFN-γ, TNF-α, IL-2, IL-4, IL-6, IL-13) in duodenum (**b**), jejunum (**c**), ileum (**d**), and colon (**e**) of pregnant mice (*n* = 6). * *p* < 0.05, ** *p* < 0.01. Letter casing represents the comparison of different segments at the same age by one-way ANOVA. The lowercase letters represent the comparison of different segments of the control groups at the same age, and the uppercase letters represent the comparison of different segments of the stress groups at the same age. The same letter indicates that the difference is not significant (*p* > 0.05), while different letters indicate that the difference is significant (*p* ≤ 0.05). CD: Duodenum of control group; CJ: Jejunum of control group; CI: Ileum of control group; CC: Colon of control group; SD: Duodenum of restraint stress group; SJ: Jejunum of restraint stress group; SI: Ileum of restraint stress group; SC: Colon of restraint stress group.

**Figure 6 ijms-24-12822-f006:**
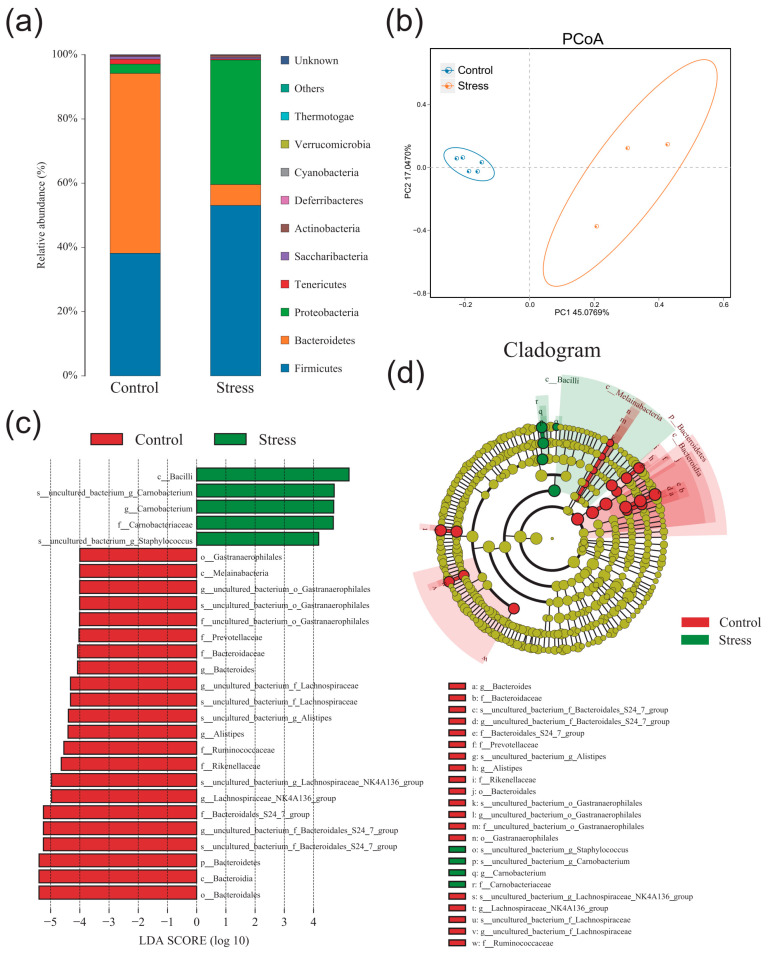
Chronic stress reduces the diversity of intestinal microorganism in the P10 group. (**a**) Species abundance histogram at the phylum level in the P10 group after chronic stress (*n* = 3–5). (**b**) PCoA analysis in the P10 group after chronic stress (*n* = 3–5). (**c**,**d**) Cladogram and LDA score in the P10 group after chronic stress (LDA score > 4.0) (*n* = 3–5).

**Figure 7 ijms-24-12822-f007:**
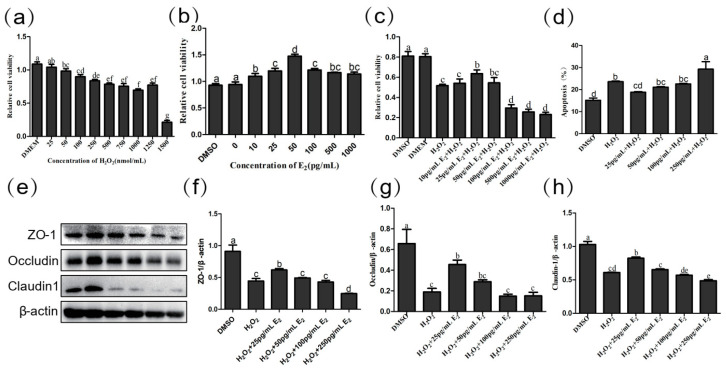
(**a**) Effect of H_2_O_2_ on IPEC cell viability (*n* = 4). (**b**) Effect of E_2_ on IPEC cell viability (*n* = 4). (**c**) Effect of E_2_ on IPEC cell viability with H_2_O_2_ treatment (*n* = 4). (**d**) Effect of E_2_ on apoptosis of IPEC cell lines in H_2_O_2_ damaged by flow cytometry analysis. (**e**) The expression of jejunum tight junction protein ZO-1 (**e**,**f**), Occludin (**e**,**g**), Claudin-1 (**e**,**h**), and β-actin (**e**) protein were examined by western blotting in H_2_O_2_-damaged IPEC cell, and relative protein levels were normalized to β-actin (*n* = 3). The same letter indicates that the difference is not significant compares with others group (*p* > 0.05), while different letters indi-cate that the difference is significant (*p* ≤ 0.05).

**Figure 8 ijms-24-12822-f008:**
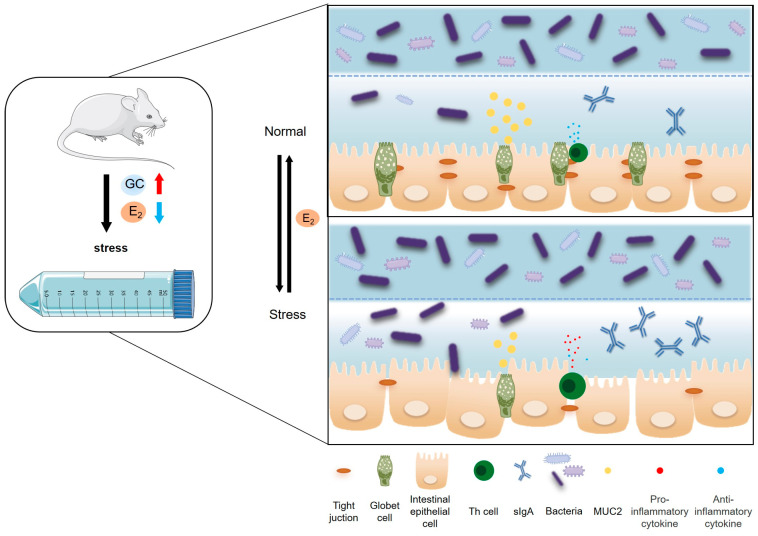
Chronic stress to female mice can significantly inhibit the activity of the HPO axis, reduce estrogen secretion, and thus disrupt the structure and function of the intestinal mucosal barrier. Firstly, it can reduce the height of intestinal villi and V/C value and increase the depth of the crypt; inhibit the proliferation of intestinal epithelial cells in female mice and promote their apoptosis; reduce the number of intestinal goblet cells and inhibit the secretion of mucin (Muc2); and reduce the expression of Claudin-1, Occludin, and ZO-1 closely connected between intestinal epithelial cells, thereby destroying the mechanical barrier of the intestinal mucosa. Secondly, it can increase the secretion of intestinal sIgA and pro-inflammatory cytokines (TNF-α, IFN-γ, IL-2, IL-6) and reduce the secretion of anti-inflammatory cytokines (IL-13, IL-4), thereby destroying the intestinal mucosal immune barrier. Finally, it can affect the composition of microorganisms in the colonic contents and reduce the abundance and diversity of microorganisms, mainly due to the decrease of Bacteroides and the increase of *Proteocytic*. In addition, when the E_2_ concentration reaches 25 pg/mL, it can improve the proliferative activity and tight connection integrity of intestinal epithelial cells, and it has a positive effect on maintaining the integrity of the intestinal epithelial barrier. (GC: glucocorticoids).

**Table 1 ijms-24-12822-t001:** Primers of target genes and reference gene.

Gene	Forward	Reverse
Muc-2	5′-CTGACCAAGAGCGAACACAA-3′	5′-CATGACTGGAAGCAACTGGA-3′
Claudin-1	5′-AGGTCTGGCGACATTAGTGG-3′	5′-TGGTGTTGGGTAAGAGGTTG-3′
Occludin	5′-CTTTGGCTACGGAGGTGGCTAT-3′	5′-CTTTGGCTGCTCTTGGGTCTG-3′
ZO-1	5′-GCATGTAGACCCAGCAAAGG-3′	5′-GGTTTTGTCTCATCATTTCCTCA-3′
β-actin	5′-TGCTGTCCCTGTATGCCTCTG-3′	5′-TTGATGTCACGCACGATTTCC-3′

## Data Availability

All relevant data are within the manuscript.

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
