# Peer review of "Chronic Stress That Changed Intestinal Permeability and Induced Inflammation Was Restored by Estrogen"

_ijms, 2023, doi:10.3390/ijms241612822_

Round 1

Reviewer 1 Report

The manuscript entitled “Chronic stress which changed the intestinal permeability and induced inflammation was restored by estrogen” is significant in this field of interest. The manuscript is well-structured with enough data. However, this manuscript has minor issues needed to be addressed. Thus I recommend this manuscript for minor revision. 

 Ø  The manuscript must undergo a typographical error check.

Ø  Revise section 4.1 to enhance clarity, especially concerning the grouping of animals.

Ø  Apply all abbreviations after citing their full names upon first occurrence in the manuscript.

Ø  Figure 6 (a-k) is too small and unclear to read, so move it to supplementary material, retaining only figures b, c, and d in the main manuscript.

Ø  Split Figure 4 into two separate figures: one containing Figure 4a and the other containing Figure 4b-f.

Author Response

Dear Reviewer,

Thank you for your valuable feedback on our manuscript.  We have carefully addressed your suggestions and made necessary revisions to improve the paper's quality. We appreciate the time and effort you put into reviewing our work. We have carefully considered your suggestions and made the necessary changes to improve the quality of the paper.
Please refer to the attached revised manuscript for further evaluation.

Thank you again for your time and consideration.

Best regards

Yuanyuan Li

Reviewer 2 Report

This is an interesting article trying to investigate how chronic stress affects intestinal permeability in pregnant mice. The findings are of importance as well as the number of experiments used herein. The use of english language needs substantial revision as there are many grammar and syntactical errors, and the presentation of the results has to be improved in order to be better understandable to the readers. Please find more specific comments below:

Abstract

Please define NE, CORT, E2. Also, there are many abbreviations throughout the text that have not been explained ie IBD, ICR etc. please revise them accordingly.

Line 18. something is wrong with the punctuation marks

19 found not find.

20 for intestinal what?

Introduction

Introduction is poor you should enrich it with more recent bibliography and findings

25 underd?

Please rephrase the first introduction sentence as it sounds stereotypical.

55 discuss not discusses

Methods

n=6 you mean 6 mice in each category, please be more specific on the number of animals used, for example mention it in each figure

380. The samples were collected. What samples?

477 endocrine system?

Results

The results should be better described with more details on what you compare, for example compared to non pregnant control? Also when you say decrease or increase you should mention compare to what, it decreased compared to P1? Please do the comparisons in every sentence in order to be more clear to the reader. For example in line 90 “The CD also decreased (P=0-0.006)”, when and compared to what? During pregnancy?

Also in the figures the letters that you use for statistical comparisons are confusing and sometimes not very clear inside the figures. Please give better explanations on the use of letters inside figures or change the way you present statistical differences.

Fig 1a you report that weight loss was increased in control P20 mice? Please give a possible explanation on why this happened.

In fig 3 PCNA is a and caspase b? in text you state the opposite. Also what do you mean the rate of PCNA?

In results 2.4 you report that sIgA has antiinflammatory effect and that stress promotes inlammation and sIgA. Please rephrase.

175 Camobacteriaccae (Fig.6(f)) you mean 6g?

Discussion

Please remove first paragraph which is from the template

245 to be depression?

295 the distribution

353 which study? Add reference

The last paragraph of discussion needs to be rewritten. It is not clear what you have done and what was already done in other studies. You mix bibliography with your present results which is confusing. It is better that the last paragraph either summarizes only your results and the strengths and limitations of your study or either discuss other studies and trying to make a generalization.

Also in discussion you should discuss more on other similar studies and compare their findings with your results. Even if there are not exact same studies you should comment on other findings in this specific scientific field.

extensive editing required

Author Response

Dear Reviewer,

Thank you for your valuable feedback on our manuscript. We have carefully addressed your suggestions and made necessary revisions to improve the paper's quality. Below are the answers to some of the questions, and the other minor errors that needed correction have been appropriately addressed in the manuscript.

In Results

Question 14: In fig 3 PCNA is a and caspase b? in text you state the opposite. Also what do you mean the rate of PCNA?

Answer: Firstly, at gestation day 20, some mice may already be in the peripartum period, while others may have completed parturition. For mice that have not given birth, the stress of impending parturition may lead to reduced appetite, resulting in significant fluctuations in body weight in the later stages of pregnancy. Secondly, in the first set of results, it can be observed that during late gestation, the levels of CORT and NE hormones were significantly higher compared to the NP group. Weight loss due to hormonal and metabolic changes is also a possible contributing factor. Lastly, during this period, mother mice require additional energy to support embryonic and uterine development, which could lead to body weight reduction in the later stages of pregnancy.

Question 16: In results 2.4 you report that sIgA has antiinflammatory effect and that stress promotes inlammation and sIgA. Please rephrase.

Answer: I apologize for the confusion.  I have made the necessary modifications in the manuscript to address this issue.  Please refer to the revised version in the attached document for the accurate and updated information.

Please find the attached revised manuscript for your perusal. We hope that these modifications address your concerns and contribute to enhancing the overall quality of the paper.

Thank you once again for your valuable input, and we look forward to your feedback on the revised version.

Best regards.

Yuanyuan Li

Round 2

Reviewer 2 Report

Most of my comments have been addressed. Please add the exact number of animals that you used in each group in more detail in the Methods and maybe in the footnote of each figure.

Minor editing of English language required

Author Response

Dear Reviewer,

We carefully considered your recommendations, and we have made significant revisions to address the issues raised.    Firstly, we have made the necessary adjustments to the animal numbers and clarified specific aspects of our study to enhance its comprehensibility.    Moreover, we have also taken steps to improve the language and overall clarity of the manuscript.

Your time and expertise in reviewing our work are greatly appreciated, and your contributions have undoubtedly enriched the research.    We believe that the revised manuscript now provides a more robust and accurate representation of our findings.

Once again, thank you for your valuable input.